# Outcome measures for children with speech sound disorder: an umbrella review protocol

Sam Harding ![ORCID] ,[1,2] Sam Burr,[1] Joanne Cleland,[3] Helen Stringer,[4] Yvonne Wren ![ORCID] [1,5]

[1]Bristol Speech and Language Therapy Research Unit, North Bristol NHS Trust, Westbury on Trym, UK
[2]Department of Research and Innovation, North Bristol NHS Trust, Westbury on Trym, UK
[3]Department of Psychological Sciences and Health, University of Strathclyde, Glasgow, UK
[4]School of Education, Communication and Language Sciences, Newcastle University, Newcastle upon Tyne, UK
[5]Bristol Dental School, University of Bristol, Bristol, UK

**Correspondence to**
Dr Sam Harding;
sharding.jb@gmail.com

## ABSTRACT

**Introduction** Speech sound disorder (SSD) describes a 'persistent difficulty with speech sound production that interferes with speech intelligibility or prevents verbal communication'. There is a need to establish which care pathways are most effective and efficient for children with SSD. Comparison of care pathways requires clearly defined, evidence-based interventions and agreement on how to measure the outcomes. At present, no list of assessments, interventions or outcomes exists.
The objective of this paper is to provide a rigorous and detailed protocol for an umbrella review of assessments, interventions and outcomes that target SSD in children. The protocol details the development of a search strategy and trial of an extraction tool.

**Methods and analyses** The umbrella review has been registered with PROSPERO (CRD42022316284). Papers included can use a review methodology of any sort but must include children of any age, with an SSD of unknown origin. In accordance with the Joanna Briggs Institute scoping review methods guidelines, an initial search of the Ovid Emcare and Ovid Medline databases was conducted. Following this, a final search strategy for these databases were produced. A draft extraction form was developed.

**Ethics and dissemination** Ethical approval is not needed for an umbrella review protocol. Following the systematic development of an initial search strategy and extraction form, an umbrella review of this topic can take place. Dissemination of findings will be through peer-reviewed publications, social media, and patient and public engagement.

## INTRODUCTION
### Terminology and prevalence

Speech sound disorder (SSD) describes a 'persistent difficulty with speech sound production that interferes with speech intelligibility or prevents verbal communication'.[1] Prevalence is high, with estimates of 3.6% in children aged 4–8 years,[2–4] and upwards of 76000 children referred to National Health Service (NHS) Speech and Language Therapy (SLT) services annually.[5] While a minority of children with SSD have clear aetiology (eg, cleft palate, cerebral palsy and hearing impairment), in most

---

### STRENGTHS AND LIMITATIONS OF THIS STUDY

⇒ The number of systematic reviews produced to inform healthcare has rapidly increased in recent years, leading to healthcare decision makers needing to source multiple articles. The proposed umbrella review is designed to collate existing systematic reviews.
⇒ This protocol follows the Preferred Reporting Items for Systematic Review and Meta-Analysis Protocols guidelines.
⇒ This protocol is also guided by the COS-STAP checklist due to its aim of defining a core outcome set.
⇒ Electronic databases in languages other than English will not be searched. This may cause language bias.

cases, SSD has no identifiable cause, and the evidence for intervention for this group is limited. Untreated, the impact of SSD is far reaching, leading to poor outcomes in education, employment and mental health.[6–9] NHS SLT is provided to children with SSD via a range of care pathways, typically defined by resource constraints, rather than robust evidence. Vanhaecht et al[10] defined a care pathway as 'a complex intervention for the mutual decision-making and organisation of care processes for a well-defined group of patients during a well-defined period'.[10] Care pathways aim to improve care, outcomes and patient satisfaction while also optimising the use of resources. Carepathway examples include total hip replacement and palliative care.

### Implications for clinical practice

There is a need to establish which care pathways are most effective and efficient for children with SSD. Comparison of care pathways requires both clearly defined, evidence-based interventions and agreement on how best to measure the outcomes of these interventions for children with SSD. However, a review of existing case notes of children treated for SSD were found to be too incomplete to compare

BMJ

## Box 1 Full search strategy for Medline

1. (child* or youth* or boy* or girl* or juvenil* or teenage* or adolescen* or "young person*" or "young people*" or toddler* or infan* or baby or babies).mp.
2. Child/ or Adolescent/ or Infant/ or Infant, Newborn/
3. 1 or 2
4. (phon* or speech or speech disorder* or speech impairment* or speech sound disorder* or speech sound difficult* or speech-sound* or speech retard* or speech delay* or speech disabilit* or speech handicap* or speech problem* or childhood apraxia of speech or apraxia of speech or developmental verbal dyspraxia or verbal dyspraxia or dyspraxia or articulat*).ti,ab.
5. exp Speech Sound Disorder/
6. 4 or 5
7. ("clinical service*" or "therap* service*" or NHS or "social care" or "social service*" or school* or education* or nurser* or "early year*" or preschool* or pre-school* or college* or universit*).mp.
8. Schools/ or Universities/ or Nurseries, Infant/ or Child, Preschool/ or Social Support/
9. 7 or 8
10. (exp META-ANALYSIS AS TOPIC/ or ("meta analy*" or "metaanaly*").ti,ab. or META-ANALYSIS/ or (systematic adj1 (review*1one or over-view*1)).ti,ab. or exp REVIEW LITERATURE AS TOPIC/ or (cochrane or embase or psychlit or psyclit or psychinfo or psycinfo or cinahl or cinhal or "science citation index" or bids or cancerlit).ab. or ("reference list*" or bibliograph* or hand-search* or "relevant journals" or "manual search*").ab. or (("selection criter*" or "data extraction").ab. and exp REVIEW/)) not ((ANIMALS/ not (ANIMALS/ and exp HUMANS/)) and (COMMENT/ or LETTER/ or EDITORIAL/ or (letter* or comment*1one or editorial*1).ti,ab.))
11. 3 and 6 and 9 and 10
12. 11
13. limit 12 to (english language and yr="2010 -Current")

pathways.[11] Additionally, it was found that preintervention and postintervention data and variables recorded in the clinical case notes varied significantly between and within SLT services, thereby negating comparison. As a first step to determining which care pathways are most effective and efficient, this umbrella review will ask which assessment and outcome measures are commonly employed with children with SSD. Moreover, Morgan and Wren suggest that there is a need for agreement on a national core outcome set for SSD.[12] A national consultation in 2018 identified the need to collect consistent data and recommended that NHS England supports providers to collect data on the quality and outcomes of interventions (recommendation 4.5, p. 29).[13]

### Reviews to date

Evidence from systematic reviews and trials has shown that intervention is effective for the majority of children with SSD and that these children do not make progress without intervention.[14 15] However, studies have typically employed intervention protocols that are intense and are difficult to replicate in NHS SLT services.[16, 17 16–18] Moreover, unlike research studies, clinical intervention takes place within care pathways that vary in terms of timing of intervention (preschool and school age), delivery

by speech and language therapists (SLTs) or assistants, number, frequency and duration of sessions, and involvement of parents or education staff, as well as the assessments and outcome measures used.

Previous research has identified that functional goals such as independence and improved social interaction are of greatest importance to parents,[18] while children listed improved speech alongside improved behaviour, schoolwork and skill at sports as well as making friends as important goals.[19 20] Preferred outcomes for preschool children with SSD among SLTs have been identified as: intelligibility, social interaction and participation.[21] However, although a list of assessment tools (ie, the tools used to obtain a speech sample) was compiled as part of this work,[21] it did not consider which specific analysis (eg, percentage consonants correct, consonant inventory and error pattern analysis) is preferred to measure the primary outcome or how this relates to functional domains (eg, social interaction, participation and inclusion).

As can be seen from the number of reviews included in this introduction, evidence syntheses undertaken using this type of methodology are increasing in frequency in published literature. They provide a rigorous and transparent knowledge base for translating clinical research into decisions and, as such, are 'go to' documents to advise healthcare service construction and evaluation. An overarching review that combines previous reviews is needed for clinicians and researchers to consolidate what we know about interventions in SSD to date. Umbrella reviews are reviews of previously published scoping reviews, systematic reviews or meta-analyses. They aim to collate and represent one of the highest levels of evidence synthesis currently available, undertaking at least in part the historical role of the systematic review.[22 23] It is in recognition of the number of potential reviews previously undertaken in the field of childhood SSD that the proposed review being outlined in the current protocol used an umbrella methodology.

### Review objective

The objective of the proposed umbrella review is to collate the tools used for initial and baseline assessment, intervention and outcome measurement with children with SSD in speech and language therapy.

### Review question

What assessment, interventions and outcomes are reported for children with SSD in health, social care and education?

## METHODS AND ANALYSIS

The review will be conducted in accordance with the Joanna Briggs Institute (JBI) methodology for umbrella reviews,[24] with an addition relating to quality appraisal where the Assessment of Multiple Systematic Reviews (AMSTAR) tool will be used.[25] This is a literature review, and therefore, ethical approval is not required. The

**Table 1** Inclusion and exclusion criteria

| Inclusion criteria | Exclusion criteria |
|---|---|
| ► Children of any age.<br>► Children with SSD of unknown origin including:<br>  – Childhood apraxia of speech/ developmental verbal dyspraxia.<br>► Articulation disorders.<br>► Phonological disorders of all types. | ► Children with SSDs associated with a biomedical condition for example:<br>  – SSD associated with cleft lip and/or palate.<br>  – Cerebral palsy.<br>  – Traumatic brain injury.<br>► Reviews not written in English.<br>► Reviews that report outcomes for adults.<br>► Reviews of studies with no reported assessments or outcomes from interventions for SSD. |

SSD, speech sound disorder.

umbrella review has been registered with PROSPERO (CRD42022316284).

### Patient and public involvement
None.

### Eligibility criteria
In line with the JBI guidance, the eligibility for included studies will be outlined according to population, phenomena of interest and context of data.[24] As this will be an umbrella review, the only papers retained for inclusion will be reviews. This can include any type of review, for example, systematic reviews of effectiveness, mixed methods, qualitative and scoping reviews.[26]

### Population
The included population is children of any age (from birth up to 18 years) with a diagnosis of SSD. Studies will not be excluded if the interventions include additional intervention targets (eg, for receptive language). Children whose speech needs are associated with a biomedical condition with a known association with communication, such as sensorineural deafness, autistic spectrum condition or cleft lip and palate and neurological conditions (eg, cerebral palsy) affecting speech output, will be excluded.

### Phenomena of interest (concept)
To be included in the umbrella review, studies must assess children or the outcomes of intervention for children with SSD. This can include articulation disorder, childhood apraxia of speech (formerly known as developmental verbal dyspraxia) or phonological disorders/delay. It will exclude children with a known cause for their SSD, such as those with identified genetic or chromosomal anomalies, and congenital or acquired neurological conditions.

### Context
The context for included reviews will be open in that it will consider reviews that retain studies taking place in any setting (eg, home, clinic and nursery) and geographical location.

### Information sources
As the aim of this umbrella review is to provide a long list of assessments, interventions, outcomes and outcome tools (measures) used in the evaluation of SSD in children, it will not exclude relevant studies on account

of their review methodology. However, to maintain a minimum standard of research quality, included reviews will have been published within peer-reviewed journals. To locate papers with this minimum quality that have been subject to peer review, grey literature will be excluded.

**Table 2** Extraction form

| Data charting |
|---|
| Evidence source details and characteristics |
| Citation details (reference) |
| Study design/type of review |
| Country of origin of the review |
| Number of articles included in review<br>Primary research question<br>Secondary research question(s)<br>PICO (Participant, Intervention, Comparision, Outcome)/PCC (Population, Concept, Context) criteria<br>Setting/context<br>Included study designs<br>Inclusion/exclusion |
| Demographic items<br>► Age<br>► Biological sex<br>► Diversity characteristics<br>► Setting<br>► Comorbidity<br>► SSD subtype |
| Interventions |
| Intervention type<br>Intervention method<br>Intervention delivered by<br>Service delivery framework<br>Service delivery format(s) |
| Therapeutic content/therapeutic dosage<br>► Dose<br>► Frequency<br>► Method |
| Assessments |
| Outcomes |
| Measurement instruments |
| Analysis performed |
| Conclusions drawn |

| Table 3 | Presentation of overarching review information | | | | |
|---|---|---|---|---|---|
| **Overarching study information and population** | | | | | |
| **Reference (country)** | **Type of study** | **Aims (as relevant to the review)** | **Number of studies included (PICO/PPC)** | **No children (biographical information)** | **Age range at baseline** |
| - | | | | | |

The complete search will include Ovid Medline, Ovid Embase Cumulative Index to Nursing and Allied Health Literature (CINAHL), Psychinfo and Cochrane. These databases have been selected because they cover a broad range of journals pertaining to medicine, psychology (including child development) and the allied health professions.

In addition to these standard journal databases other platforms will be integrated including Campbell collaboration, COnsensus-based Standards for the selection of health Measurement INstruments, Figshare, JBI, Open Science Framework, PROSPERO and Speechbite. Due to a limitation in resources, included studies will be in English. In order to included literature hopefully relevant to current speech and language therapy practice, the search will have a minimum publication year of 2010 (1 January 2010). Where a potentially relevant review article cannot be retrieved, direct contact with the study authors will be made.

### Search strategy

In accordance with JBI protocol development guidance an initial limited search of two databases was conducted prior to the full search being carried out.[24] Initially, a set of key terms was developed by the first author, in consultation with two independent subject experts with significant postdoctoral research experience in the area. These terms were used for the initial limited search of Ovid Medline and Ovid Embase to identify articles on the topic. With the support of a clinical librarian, the text words contained in the articles and abstracts of relevant articles and the index terms used to describe the articles were used to develop a full search strategy for Medline. Box 1 presents the full search strategy for Medline. When completing the database search for the full review, keywords and index terms will be adapted for each selected database as appropriate. The reference list of all included sources of evidence will be screened for additional studies.

### Study/source of evidence selection

Following the search, all identified citations will be collated and uploaded into Endnote and duplicates removed. Inclusion and exclusion criteria are presented in table 1. Titles of studies that are clearly unrelated to the population and concept of the umbrella review will also be removed. Two reviewers will independently review 100% of the remaining abstracts against the inclusion criteria as stated. They will meet to compare their selection of articles. Where disagreement is present, the two reviewers will meet to discuss and if consensus is not achieved a third reviewer will be included in the discussion. Once all abstracts have been reviewed, potentially relevant sources for full-text review will then be retrieved in full and imported into the Rayyan.ai system for the systematic review management.[27] The two reviewers will examine all selected papers independently at full text level with regular consensus meetings. Reasons for the exclusion of sources at full text will be recorded and reported in the umbrella review. Any disagreements that arise between the reviewers at each stage of the selection process will be resolved through either discussion or with an additional reviewer/s. The results of the search and the study inclusion process will be reported in full in the final umbrella review and presented in a Preferred Reporting Items for Systematic Reviews and Meta-analyses extension for umbrella review flow diagram.[28]

Following final selection and retention of review articles, critical appraisal will be undertaken using the AMSTAR tool.[25] This tool is selected as it is designed to critically appraise systematic reviews that include randomised or non-randomised studies of healthcare interventions, or both. Two reviewers will individually appraise each study, with consensus meetings to confirm ratings. As with the study inclusion process, if consensus cannot be met, a third reviewer will be consulted.

### Data extraction

Data from the retained reviews will be identified using a researcher-developed extraction form. This form was adapted from guidance provided by the JBI Reviewer's Manual in order to meet the specific requirements of the proposed review.[26] The form was piloted by two independent reviewers on two relevant studies identified from the initial limited search. A final draft was agreed following a consensus meeting between the two reviewers. The final draft was amended to include specific details about the population and concept as relevant to the aims of this

| Table 4 | Presentation of outcomes and measurement instruments | | | | |
|---|---|---|---|---|---|
| **Speech interventions** | **Speech assessments** | **Speech outcomes** | **Measurement instruments** | **Analysis performed** | **Conclusions drawn** |
| - | | | | | |

umbrella review. Study design, population details include age, assessment, intervention, comparison, outcome and context will be extracted. Where these are not presented for each paper in a retained review, the source paper will be obtained, and the data extracted from that. The data extraction tool (table 2) will be revised if necessary, during the process of extracting data from each included information source.

## Analysis of the evidence

The results of each included review paper will not be independently reported as this study is not being conducted within a systematic review methodology.[26] However, as a broad overview of study quality has been included, a brief synthesis of overall study findings will be reported narratively.

The principles from Grading of Recommendations Assessment, Development, and Evaluation (GRADE) will be used for an overall assessment of the quality of evidence for each intervention or phenomena of interest. The GRADE concept is based on an assessment of the following criteria: quality of primary studies, design of primary studies, consistency and directness.[29]

## Presentation of the results

The overall study information with concept and context data will be presented in tabular form with a corresponding narrative summary for each section (tables 3 and 4). The findings from the quality appraisal (AMSTAR) will be discussed narratively, with tables summarising reviewer appraisal ratings. As the presentation of data is an iterative process dependent on study findings,[24] these presentation approaches may be further refined at review stage according to the content of the findings.

## DISSEMINATION

This protocol has described the initial limited search process, the development of a usable extraction tool, as well as an overview of how evidence will be analysed and presented. The next stage will be to conduct the full review and report on the findings as in accordance with this protocol.

**Contributors** SH devised the search strategy and the data extraction form. SH also wrote the first full draft of the manuscript. SB, JC, HS and YW reviewed, revised and commented on the developed search strategy. All authors reviewed and agreed final manuscript.

**Funding** The authors of this paper are holders of a Research for Patient Benefit (RfPB) award (NIHR202766) and are funded in partnership by the NIHR for this research project.

**Disclaimer** The views expressed in this publication are those of the author(s) and not necessarily those of theNational Institute for Health and Care Research, National Health Service or the UK Department of Health and Social Care.

**Competing interests** None declared.

**Patient and public involvement** Patients and/or the public were not involved in the design, or conduct, or reporting, or dissemination plans of this research.

**Patient consent for publication** Not applicable.

**Ethics approval** Not applicable.

**Provenance and peer review** Not commissioned; externally peer reviewed.

**Data availability statement** No data are available. Review protocol.

**ORCID iDs**
Sam Harding http://orcid.org/0000-0002-5870-2094
Yvonne Wren http://orcid.org/0000-0002-1575-453X

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
