## [Reviewer comments · BMJ Open]

ARTICLE DETAILS

TITLE (PROVISIONAL)	Outcome measures for Children with Speech Sound Disorder: An Umbrella Review Protocol
AUTHORS	Harding, Sam; Burr, Sam; Cleland, Joanne; Stringer, Helen; Wren, Yvonne

VERSION 1 – REVIEW

REVIEWER	Morgan, Angela Murdoch Children's Research Institute, Melbourne, Victoria, Australia
REVIEW RETURNED	17-Oct-2022

GENERAL COMMENTS	Thankyou for the opportunity to review this important paper. Is the inclusion criteria missing childhood apraxia of speech? It is noted that dysarthria or acquired disorders are expressly mentioned. Think CAS should also be noted? Important to acknowledge Cochrane reviews by Pennington et al. and Morgan et al. also? Around dysarthria and apraxia. The exclusion criteria seems at odds with the inclusion criteria around children with biomedical issues? Or is there a distinction that you are not including children with congenital issues and only acquired? Cerebral Palsy has a genetic bases now also. Will children with genetic based speech conditions not be included? I think that is the case? Or maybe the wording of the inclusion criteria needs to be altered - perhaps this should more go under exclusion? Children diagnosed with an SSD subtype other than: o SSD associated with cleft lip and/or palateo Dysarthriao acquired SSD If motor speech disorders are being excluded - more of a justification is required. It was not entirely clear - apraxia was mentioned in the search terms...
--

REVIEWER	DeVeney, Shari L University of Nebraska Omaha
REVIEW RETURNED	05-Dec-2022

GENERAL COMMENTS	Overview: Thank you for the opportunity to appraise this interesting and informative review protocol for BMJ Open. The planned umbrella review involves an investigation of speech sound disorder intervention outcomes and the methodology presented for determining these outcomes appears well-thought through and sound. However, the authors are encouraged to enhance the introduction in such a way as to better establish the rationale for undertaking such a project and refine the research aims for clarity.
---

	The following recommendations are offered to strengthen the manuscript overall: The authors should consider defining “care pathways” (p. 4) and providing examples of existing pathways to enhance clarity for the BMJ Open readership. Not all readers will be familiar with such terminology. Further, please provide more context for the statement “...varied significantly between and within SLT services...” (p. 4). In what was were there evidence of significant variability? Please expand. The authors are encouraged to take service delivery format under consideration as well as this can vary substantially between and within SLT services provided to children with speech sound disorder (SSD). Noting “frequency and duration of sessions” can be included and more holistically referenced as intervention “dosage intensity” (see descriptions of dosage intensity of SSD intervention: Warren, Fey, & Yoder in 2007; Baker & Williams in 2011). To enhance interpretation and clarity, spell out all acronyms at first use throughout the manuscript (e.g., AMSTAR = assessment of multiple systematic reviews). For research aims, please specify if the researcher intend to find assessment tools used in the initial evaluation at entry into SLT services or assessment tools used to determine when dismissal is appropriate (i.e., those used for outcome assessment)? The assessment tools used for each of these two purposes may vary. Please be explicit in sharing the intent of the review. For the inclusion/exclusion criteria presented, please provide the upper bounds of what would be considered ‘childhood’ as this may vary according to source. What do the authors mean by “children of any age?” This reviewer encourages them to specific exact age range for clarity and for easy of replication purposes. The authors are encouraged to better account for and acknowledge the vast body of work that has already been completed in the area of pediatric SSD intervention regarding the variables of interest (e.g., Baker & McLeod, 2011; McLeod & Baker, 2014; Skahan, Watson, & Lof, 2007, etc.) Finally, the authors are encouraged to include an explanation regarding how an ‘umbrella’ review differs from a scoping review so that the aim of the review protocol is firmly established. Thank you again for the opportunity to assess this manuscript. As a researcher in this area, this reviewer is quite interested in the outcome of this proposed umbrella review.
--	--

VERSION 1 – AUTHOR RESPONSE

Reviewer: 1

Thankyou for the opportunity to review this important paper. Is the inclusion criteria missing childhood apraxia of speech? It is noted that dysarthria or acquired disorders are expressly mentioned. Think CAS should also be noted?

- Childhood apraxia of speech has been added to the inclusion criteria, but please feel reassured that it was already included in the search string and in the definition of the concept being investigated.

Important to acknowledge Cochrane reviews by Pennington et al. and Morgan et al. also? Around dysarthria and apraxia. The exclusion criteria seems at odds with the inclusion criteria around children with biomedical issues? Or is there a distinction that you are not including children with congenital issues and only acquired? Cerebral Palsy has a genetic bases now also. Will children with genetic based speech conditions not be included? I think that is the case? Or maybe the wording of the inclusion criteria needs to be altered - perhaps this should more go under exclusion? Children diagnosed with an SSD subtype other than: o SSD associated with cleft lip and/or palate o Dysarthria o acquired SSD

- Thank you for pointing out this oversight. It was our intention to include all children with SSD of (as yet) unknown origin, and therefore we include CAS, but exclude dysarthria as associated with, for example, CP. Exclusion criteria have been added to in order to clarify this issue.

If motor speech disorders are being excluded - more of a justification is required. It was not entirely clear - apraxia was mentioned in the search terms...

- Amended sentence under Phenomena of interest: "*This can include articulation disorder, childhood apraxia of speech, (formerly known as developmental verbal dyspraxia) or phonological disorders/delay. It will exclude children with congenital or genetic disorders and **diagnosed neurological disorders such as dysarthria and acquired speech sound disorders.***"

Reviewer: 2

Overview: Thank you for the opportunity to appraise this interesting and informative review protocol for BMJ Open. The planned umbrella review involves an investigation of speech sound disorder intervention outcomes and the methodology presented for determining these outcomes appears well-thought through and sound. However, the authors are encouraged to enhance the introduction in such a way as to better establish the rationale for undertaking such a project and refine the research aims for clarity. The following recommendations are offered to strengthen the manuscript overall:

- Thank you for this comment. We have added a section justifying the use of an umbrella review as a way of to consolidate what we know about interventions in SSD to date "As can be seen from the number of reviews included in this introduction, evidence syntheses such undertaken using this type of methodology are increasing in frequency in published literature. They provide a rigorous and transparent knowledge base for translating clinical research into decisions, and as such are 'go to' documents to advise healthcare service construction and evaluation. An overarching review which combines previous reviews is needed for clinicians and researchers to consolidate what we know about interventions in SSD to date. Umbrella reviews are reviews of previously published scoping reviews, systematic reviews or meta-analyses. They aim to collate and represent one of the highest levels of evidence synthesis currently available, undertaking at least in part the historical role of the systematic review. It is in recognition of the number of potential reviews previously undertaken in the field of

childhood SSD that the proposed review being outlined in the current protocol used an Umbrella methodology. "

The authors should consider defining "care pathways" (p. 4) and providing examples of existing pathways to enhance clarity for the BMJ Open readership. Not all readers will be familiar with such terminology.

- The following additional information has been added to this section: "Vanhaecht et al defined a care pathway as "a complex intervention for the mutual decision-making and organisation of care processes for a well-defined group of patients during a well-defined period." Care pathways aim to improve care, outcomes and patient satisfaction while also optimising the use of resources.[10] Care pathway examples include total hip replacement and palliative care."

Further, please provide more context for the statement "...varied significantly between and within SLT services..." (p. 4). In what was there evidence of significant variability? Please expand.

- The sentence has been amended to read "Additionally, it was found that pre- and post-intervention data and variables recorded in the clinical case notes varied significantly between and within SLT services, thereby negating comparison."

The authors are encouraged to take service delivery format under consideration as well as this can vary substantially between and within SLT services provided to children with speech sound disorder (SSD).

- This has been added to the data extraction form.

Noting "frequency and duration of sessions" can be included and more holistically referenced as intervention "dosage intensity" (see descriptions of dosage intensity of SSD intervention: Warren, Fey, & Yoder in 2007; Baker & Williams in 2011).

- Thank you for this description, it has been added to the data extraction sheet and the references are greatly appreciated.

To enhance interpretation and clarity, spell out all acronyms at first use throughout the manuscript (e.g., AMSTAR = assessment of multiple systematic reviews).

- Thank you for spotting this oversight. We have made this amendment

For research aims, please specify if the researchers intend to find assessment tools used in the initial evaluation at entry into SLT services or assessment tools used to determine when dismissal is appropriate (i.e., those used for outcome assessment)? The assessment tools used for each of these two purposes may vary. Please be explicit in sharing the intent of the review.

- Amended text: "The objective of the proposed umbrella review is to collate the tools used for initial and baseline assessment, intervention and outcome measurement with children with SSD in speech and language therapy."

For the inclusion/exclusion criteria presented, please provide the upper bounds of what would be considered 'childhood' as this may vary according to source. What do the authors mean by "children of any age?" This reviewer encourages them to specify exact age range for clarity and for easy of replication purposes.

- Thank you for highlighting this. We have now added a lower and upper age limit (from birth up to 18 years)

The authors are encouraged to better account for and acknowledge the vast body of work that has already been completed in the area of pediatric SSD intervention regarding the variables of interest (e.g., Baker & McLeod, 2011; McLeod & Baker, 2014; Skahan, Watson, & Lof, 2007, etc.)

- We have added a paragraph into the introduction of the protocol giving a much clearer overview of where umbrella reviews sit in the growing field of data synthesis methodologies. Umbrella reviews by their nature start from the premise that there is a vast number of systematic reviews. We are therefore reluctant to mention specific systematic reviews in the protocol not wishing to give them prominence prior to undertaking the review.

Finally, the authors are encouraged to include an explanation regarding how an 'umbrella' review differs from a scoping review so that the aim of the review protocol is firmly established.

- We have added a paragraph to the introduction indicating the point of this type of review methodology and hope you feel it establishes its use.